# Body Checking and Body Image Avoidance as Partial Mediators of the Relationship between Internalized Weight Bias and Body Dissatisfaction

**DOI:** 10.3390/ijerph19169785

**Published:** 2022-08-09

**Authors:** Brooke L. Bennett, Allison F. Wagner, Janet D. Latner

**Affiliations:** 1Rudd Center for Food Policy & Health, University of Connecticut, Hartford, CT 06103, USA; 2Counseling and Psychological Services, University of California San Diego, La Jolla, CA 92093, USA; 3Department of Psychology, University of Hawai‘i at Mānoa, Honolulu, HI 96822-2294, USA

**Keywords:** body dissatisfaction, body image, weight bias internalization, body checking, body image avoidance

## Abstract

Internalized weight bias is associated with body image disturbances and the development of disordered eating. The association between weight bias internalization and body dissatisfaction has proven difficult to disrupt. In order to develop more effective interventions, we must identify the behavioral targets which account for this robust association. The present study sought to examine whether body checking and body image avoidance mediate the relationship between weight bias internalization and body dissatisfaction. In total, 279 female undergraduates (*M_age_* = 20.13, *SD* = 4.10) were administered a battery of survey measures. Results demonstrated that body checking partially mediates the relationship between weight bias internalization and body dissatisfaction, *Z* = 7.42, *p* < 0.001. Body image avoidance was also found to partially mediate the relationship between weight bias internalization and body dissatisfaction, *Z* = 70.03, *p* < 0.001. These findings suggest that body checking and body image avoidance may both partially account for the association between weight bias internalization and body dissatisfaction. These findings extend the existing literature on weight bias internalization by highlighting two behavioral targets for prevention and intervention efforts. Understanding these relationships has important implications for both reducing weight bias internalization and improving body dissatisfaction.

## 1. Introduction

Internalized weight bias (IWB), or the extent to which individuals endorse and apply negative weight-based stereotypes to themselves, has a host of negative associations with general well-being and psychopathology [1,2] and specifically with disordered eating attitudes, behaviors, and body dissatisfaction [3,4,5,6]. Across studies examining links between IWB and body image, negative correlations have consistently been found across both clinical and community samples representing diverse body sizes, such that higher IWB is associated with poorer body image and greater body dissatisfaction, even while statistically controlling for variables such as body mass index (BMI) and depression [2]. Similarly, IWB consistently has strong and positive associations with presence and frequency of binge eating, independent of BMI, and moderate to strong correlations with measures of global eating pathology [2]. With estimates suggesting that 20% of US adults across weight categories report high levels of IWB [7], the scope of the problem may be far-reaching.

IWB’s strong associations with multiple facets of body image and psychopathology highlight the importance of both IWB and body image disturbance as variables for continued study. Body image disturbance is considered a core feature predictive of multiple eating disorders and eating disorder behaviors [8,9], and holds meaningful negative associations with psychological distress in general [10]. Body checking consists of the repeated checking of weight and shape, and body avoidance involves the avoidance of situations that cause one to interact with weight and shape. Body checking and avoidance are considered by cognitive behavioral researchers and clinicians to be two operationalizations of the core body image disturbance hypothesized to lead to the development and maintenance of eating disorders [11]. Despite the strong evidence linking these two behaviors to body image distress and impaired psychological functioning [12], research examining their connection to IWB is lacking. One study highlighted body image avoidance as a mediator of the relationship between IWB and disordered eating [13]. Examining body checking and body avoidance may be useful in understanding the behavioral links between IWB and body image disturbance and optimizing future behaviorally focused interventions.

Researchers and clinicians have attempted to develop interventions to interrupt IWB. Several studies have examined the effects of delivering weight stigma information as part of healthy lifestyle interventions and documented decreases in IWB, eating pathology, and body dissatisfaction [14,15]; however, the context of the intervention makes it difficult to isolate the specific effects of decreased weight stigma. Initial research suggests that group-based cognitive behavioral interventions focused on psychoeducation about weight and weight stigma and identifying and restructuring cognitive distortions about weight and body image have potential to reduce IWB and fat phobia [16]. Interventions focused on gratitude have also yielded significant decreases in IWB and increases in functionality appreciation and body satisfaction [17,18].

Though emerging literature has evidenced the possibility of intervening to reduce IWB and body image, this research could be enhanced by adding behavioral markers of body image, namely body checking and body avoidance. The current study attempted to fill this gap by examining how body checking and body avoidance relate to IWB and body image. It was hypothesized that internalized weight bias would be positively related to body dissatisfaction, such that increases in IWB would correspond with increases in body dissatisfaction, and that body checking and body avoidance would partially mediate associations between internalized weight bias and body dissatisfaction.

## 2. Materials and Methods

### 2.1. Participants and Procedures

All procedures were approved by the University of Hawai‘i at Mānoa’s Institutional Review Board. Participants were 279 female-identifying college students. Participants were recruited through introductory psychology courses and offered course credit for participation. All participants completed the informed consent and study questionnaires electronically. Approximately 16.8% of the sample (n = 47) identified as being of Hispanic, Latino/a, or Spanish origin. Participants identified primarily as White/Caucasian (30.1%), Filipino (19.4%), Japanese (15.8%), Chinese (7.5%), Korean (5.4%), Native Hawaiian (5.4%), Vietnamese (2.9%), Pacific Islander (2.5%), Black or African American (1.4%), Asian Indian (0.4%), and Not listed (8.2%). Mean age of the sample was 20.13 years (*SD* = 4.10).

### 2.2. Measures

Demographic information was collected on self-reported age, ethnicity, height, and weight. We collected this information in order to characterize the sample.

#### 2.2.1. Weight Bias Internalization Scale—Modified Scale (WBIS-M) 

The WBIS-M [19] is a 11-item self-report measure which assesses the degree to which respondents hold negative beliefs about themselves due to their weight or size. The WBIS-M is a modified version of the original Weight Bias Internalization Scale [1]; the measure was modified to be used by participants of all weight statuses. Each item is rated on a 7-point scale from “strongly disagree” to “strongly agree” (e.g., “I hate myself for being overweight.”). Higher scores indicate greater internalized bias. In the current sample, internal consistency was α = 0.94.

#### 2.2.2. Body Shape Questionnaire—8-Item Version (BSQ-8C) 

The Body Shape Questionnaire [20] was used to assess concerns about body image. Each item is rated on a 1 (never) to 6 (always) scale (e.g., “Have you been particularly self-conscious about your shape when in the company of other people?”). Higher scores on the BSQ indicate greater body dissatisfaction. The BSQ has demonstrated good test–retest reliability, concurrent and discriminant validity, and internal consistency in populations of college age women [21]. However, research suggests the 34-item version lacks sensitivity to change or intervention [22]. The reduced-item version of the BSQ [21] has demonstrated superior sensitivity [22] and, therefore, was used in the present study. In the present sample, Cronbach’s alpha was α = 0.94.

#### 2.2.3. Body Checking Questionnaire (BCQ)

The BCQ [23] is a 23-item self-report measure which assesses body checking behaviors. Each item is rated on a 5-point scale from “never” to “very often” (e.g., “I check to see if my thighs spread when I’m sitting down.”). In addition to the total score, there are also three subscales, the overall appearance subscale, the specific body parts subscale, and the idiosyncratic checking subscale. For the present study, only the total score was used. Only the total score was used to remain consistent with the other measures, which provide only total scores for each construct. Higher scores indicate greater body checking. In the current sample, internal consistency was α = 0.93.

#### 2.2.4. Body Image Avoidance Questionnaire (BIAQ)

The BIAQ [24] is a 19-item self-report measure which assesses avoidance of situations that provoke concern about physical appearance. Each item is rated on a 6-point scale from “never” to “always” (e.g., “I do not go out socially if I will be ‘checked out’.”). Higher total scores indicate greater body image avoidance. In the current sample, internal consistency was α = 0.83. 

### 2.3. Statistical Analyses

Means and standard deviations were calculated for demographic variables, the BCQ, BIAQ, BSQ, and WBIS-M. Next, bivariate correlations were conducted to examine the preliminary relationship between body checking, body image avoidance, weight bias internalization, and body dissatisfaction. Finally, two mediation analyses were conducted to answer the primary research questions: (1) examining the role of body checking on the association between weight bias internalization and body dissatisfaction and (2) examining the role of body image avoidance on the association between weight bias internalization and body dissatisfaction. Mediation was conducted using the steps described by Baron and Kenny [25] and the Process Plug-in for SPSS (International Business Machines, Armonk, USA) created by Hayes [26]. Mediation results were confirmed by estimating the indirect effect and its significance using a Sobel’s test [27]. See Figure 1 and Figure 2 for visual diagrams of the mediated relationships.

## 3. Results

First, means and standard deviations were calculated for demographic variables, the BCQ, BIAQ, BSQ, and WBIS-M (Table 1). 

### 3.1. Bivariate Correlations

Next, bivariate correlations were conducted to examine the preliminary relationship between body checking, body image avoidance, weight bias internalization, and body dissatisfaction (Table 2). Consistent with previous literature, weight bias internalization and body dissatisfaction had a significant, positive association, *r*(264) = 0.72, *p* < 0.001. Weight bias internalization was positively associated with both body checking, *r*(263) = 0.52, *p* < 0.001, and body image avoidance, *r*(261) = 0.62, *p* < 0.001. Body dissatisfaction was positively associated with both body checking, *r*(266) = 0.72, *p* < 0.001, and body image avoidance, *r*(264) = 0.69, *p* < 0.001.

### 3.2. Mediational Analyses

#### 3.2.1. Body Checking

Weight bias internalization was found to be a significant predictor of body dissatisfaction, *b* = 5.12, *t*(263) = 16.66, *p* < 0.001 (Table 3), fulfilling the first criterion for the Baron and Kenny [25] mediation model. To evaluate the second criterion, weight bias internalization was used to predict the mediator variable of body checking, *b* = 6.30, *t*(263) = 9.95, *p* < 0.001. The relationship between body checking and body dissatisfaction was then examined when controlling for weight bias internalization, and the two were significantly related, *b* = 3.37, *t*(262) = 11.37, *p* < 0.001. Lastly, the mediated relationship between weight bias internalization and body dissatisfaction was examined for a drop in predictive value when the mediator was added to the model. Partial mediation was found, *b* = 0.28, *t*(262) = 11.32, *p* < 0.001. The Sobel test was used to determine that the ab effect was significantly greater than zero, *Z* = 7.42, *p* < 0.001.

#### 3.2.2. Body Image Avoidance

Weight bias internalization was found to be a significant predictor of body dissatisfaction, *b* = 5.14, *t*(261) = 16.64, *p* < 0.001 (Table 3), fulfilling the first criterion for the Baron and Kenny [25] mediation model. To evaluate the second criterion, weight bias internalization was used to predict the mediator variable of body image avoidance, *b* = 4.74, *t*(261) = 12.88, *p* < 0.001. The relationship between body image avoidance and body dissatisfaction was then examined when controlling for weight bias internalization, and the two were significantly related, *b* = 3.31, *t*(260) = 9.43, *p* < 0.001. Lastly, the mediated relationship between weight bias internalization and body dissatisfaction was examined for a drop in predictive value when the mediator was added to the model. Partial mediation was found, *b* = 0.39, *t*(260) = 8.34, *p* < 0.001. The Sobel test was used to determine that the ab effect was significantly greater than zero, *Z* = 70.03, *p* < 0.001.

## 4. Discussion

The current study examined the interrelationships among body checking, body image avoidance, internalized weight bias, and body dissatisfaction. Consistent with existing literature that identified body checking and body image avoidance as core expressions of body dissatisfaction [28], bivariate correlations showed that more frequent body checking and body image avoidance were associated with higher levels of body dissatisfaction. Supporting the first hypothesis, results also demonstrated that greater body checking and body image avoidance were associated with stronger IWB, and that body dissatisfaction and IWB were positively correlated such that as one increases, so does the other. This is in line with existing literature’s findings that IWB is associated with body image disturbances and the development of disordered eating, specifically with disordered eating attitudes, behaviors, and body dissatisfaction [29,30].

In support of the second hypothesis, both body checking and body image avoidance partially mediated the relationship between IWB and body dissatisfaction. These findings build on previous models of body dissatisfaction by suggesting that body checking and body image avoidance may both account for some of the association between weight bias internalization and body dissatisfaction. More specifically, it is proposed that individuals who have internalized negative weight-based stereotypes may engage in behaviors that affirm and maintain these beliefs. For example, individuals engaging in body checking may be negatively assessing themselves in comparison to the internalized sociocultural beauty ideal, pushing them further away from a positive or neutral body image [31]. For individuals who engage in body image avoidance, it is proposed that they create a negative feedback loop in which they avoid situations that would cause them to interact with weight and shape due to anticipated distress, thereby both preventing themselves from testing their beliefs about this distress and exacerbating potential misperception of their bodies [32]. Cognitive behavioral models of body dissatisfaction have already established the links among body checking, body avoidance, and body dissatisfaction [28]. The present study appears to be the first to identify a pathway in which greater IWB, coupled with behaviors to either check or avoid body shape and weight, may perpetuate dissatisfaction with one’s body.

The present findings are of relevance given the high rates of both IWB and body dissatisfaction within the general population. In fact, prevalence estimates suggest that approximately 20% of adults across weight categories report high levels of IWB [7], while estimates of body dissatisfaction range between 11% and 72% for women [33]. Both IWB and body dissatisfaction have been linked to a multitude of negative health outcomes, across populations and body types [34].

Additionally, understanding the interrelationship between body checking, body image avoidance, IWB, and body dissatisfaction has important implications for reducing weight stigma. Currently, many of the existing interventions designed to reduce IWB have focused on the delivery of psychoeducation materials [14,15,16], restructuring cognitive distortions [16], and expressing gratitude for one’s body [17,18]. The findings of the present study highlight the role behavioral targets may have in maintaining body dissatisfaction. Future IWB interventions might emphasize reducing behaviors such as body checking and body image avoidance. For example, mirror or other body-focused exposure exercises could be included to reduce body image avoidance. Body checking could be targeted through self-monitoring to increase awareness of triggers and through the addition of stimulus control training to reduce engagement in body checking behaviors. Research should determine whether the addition of these targets improves the efficacy of interventions to improve body image.

There are several limitations in the present study. First, this study was cross-sectional, precluding any causal conclusions. Future research should examine these research questions longitudinally and experimentally to confirm the directionality and significance of the results. Second, this study was conducted using a sample of young, college-aged women, potentially limiting its generalizability to samples of different ages and genders. The conceptual overlap between IWB and body dissatisfaction may be higher in such samples [35,36,37]. Future research should examine these research questions in samples of people who identify as male or non-binary to see if the results are consistent. Future research should also consider testing these relationships among adolescents and older adults to examine whether body checking and body image avoidance remain relevant factors in the relationship between IWB and body dissatisfaction.

## 5. Conclusions

In summary, the present study identified a novel interrelationship between body checking, body image avoidance, internalized weight bias, and body dissatisfaction. These findings extend the existing literature on weight bias internalization by highlighting two behavioral targets for prevention and intervention efforts. Understanding these relationships has important implications for both reducing weight bias internalization and improving body dissatisfaction.

## Figures and Tables

**Figure 1 ijerph-19-09785-f001:**
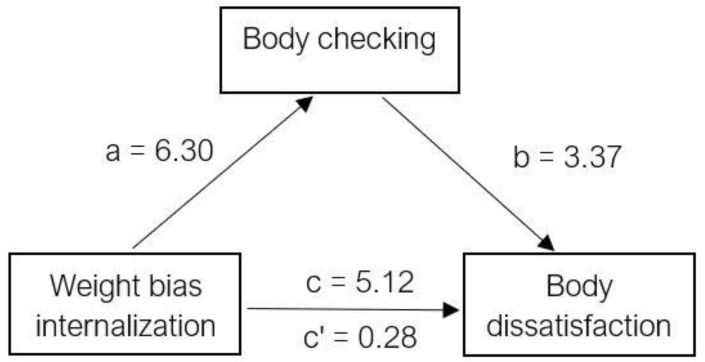
Mediated relationship between weight bias internalization and body dissatisfaction with body checking as the mediator.

**Figure 2 ijerph-19-09785-f002:**
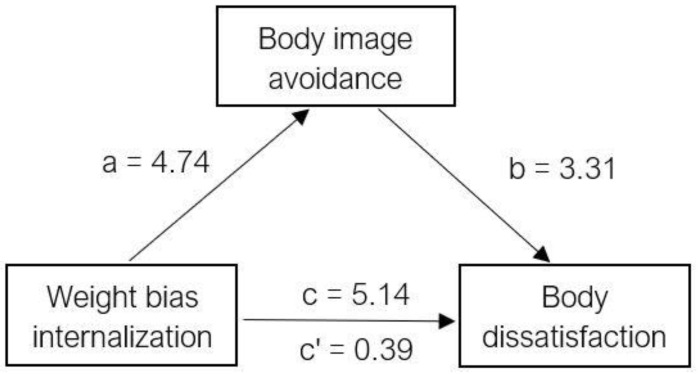
Mediated relationship between weight bias internalization and body dissatisfaction with body image avoidance as the mediator.

**Table 1 ijerph-19-09785-t001:** Means and standard deviations of study constructs.

Construct	M	SD
Body Checking Questionnaire	55.21	170.05
Body Image Avoidance Questionnaire	46.62	10.78
Weight Bias Internalization Scale—Modified	3.45	1.42
Body Shape Questionnaire—8 item	24.71	100.09

**Table 2 ijerph-19-09785-t002:** Correlations between body checking, body image avoidance, body dissatisfaction, and weight bias internalization.

	WBIS-M	BSQ	BCQ	BIAQ
WBIS-M	-	-	-	-
BSQ-8	0.72 ***	-	-	-
BCQ	0.52 ***	0.72 ***	-	-
BIAQ	0.62 ***	0.69 ***	0.64 ***	-

*** indicates significance at *p* < 0.001. Note: WBIS-M = Weight bias internalization scale—modified; BSQ-8 = Body shape questionnaire—8 item; BCQ = Body checking questionnaire; BIAQ = Body image avoidance questionnaire.

**Table 3 ijerph-19-09785-t003:** Statistics for mediation model paths.

	F	*p*	R^2^
Body checking:			
Weight bias internalization predicting body checking (a)	(1263) = 990.06	<0.001	0.27
Body checking predicting body dissatisfaction (b)	(2262) = 269.84	<0.001	0.67
Weight bias internalization predicting body dissatisfaction (c)	(1263) = 277.52	<0.001	0.51
Weight bias internalization predicting body dissatisfaction with mediation (c’)	(2262) = 269.84	<0.001	0.67
Body image avoidance:			
Weight bias internalization predicting body image avoidance (a)	(1261) = 165.80	<0.001	0.39
Body image avoidance predicting body dissatisfaction (b)	(2260) = 209.73	<0.001	0.62
Weight bias internalization predicting body dissatisfaction (c)	(1261) = 2770.03	<0.001	0.51
Weight bias internalization predicting body dissatisfaction with mediation (c’)	(2260) = 209.73	<0.001	0.62

## Data Availability

The data presented in this study are available on request from the corresponding author.

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
