# Peer review of "Body Checking and Body Image Avoidance as Partial Mediators of the Relationship between Internalized Weight Bias and Body Dissatisfaction"

_ijerph, 2022, doi:10.3390/ijerph19169785_

Round 1

Reviewer 1 Report

Discussion

The discussion section is very weak.

I suggest that references used in the introduction should not be repeated in the discussion unless very necessary. Line 192-197

In the second paragraph of the discussion no reference.

Again in the third paragraph line 216, and 219 previous references used in intro are repeated.

Again in the next paragraph references are the same as in the introduction. Line 220-232

The last paragraph of the discussion has only one reference.

This study has not discussed results with sufficient and relevant literature.

Discussion results with sufficient and fresh literature. Don’t use studies that are in the introduction section.

discussion is an important section and weakness like this is unacceptable

Reviewer 2 Report

This is a very well-written paper and my feedback is limited. Here are my comments:

--How were the demographic variables used in the analyses other than descriptives?

--The research questions in section 2.3 read as causal.

--Why did you only use the total score of the BCQ?

--The use of BMI contributes to a culture of fat phobia and weight stigma. Thus, it is concerning to me that a paper investigating weight stigma would utilize the BMI at all. It is my suggestion to eliminate it altogether.

Round 2

Reviewer 1 Report

Accepted; authors tried to improve the revisions.